

# A Mass- and Energy-Conserving Framework for Using Machine Learning to Speed Computations

Patrick Obin Sturm[1,3], Anthony S. Wexler[1,2]

[1]Air Quality Research Center, University of California, Davis, California 95616 USA
[2]Departments of Mechanical and Aerospace Engineering, Civil and Environmental Engineering, and Land, Air and Water Resources, University of California, Davis, California 95616 USA
[3]Institute of Mathematics, Technical University of Berlin, Berlin 10587 Germany

*Correspondence to*: P. Obin Sturm (posturm@ucdavis.edu)

**Abstract.** Large air quality models and large climate models simulate the physical and chemical properties of the ocean, land surface and/or atmosphere to predict atmospheric composition, energy balance, and the future of our planet. All of these models employ some form of operator splitting, also called the method of fractional steps, in their structure, which enables each physical or chemical process to be simulated in a separate operator or module within the overall model. In this structure, each of the modules calculates property changes for a fixed period of time; that is, property values are passed into the module which calculates how they change for a period of time and then returns the new property values, all in round robin between the various modules of the model. Some of these modules require the vast majority of the computer resources consumed by the entire model so increasing their computational efficiency can either improve the model's computational performance or enable more realistic physical or chemical representations in the module, or a combination of these two. Recent efforts have attempted to replace these modules with ones that use machine learning tools to memorize the input-output relationships of the most time-consuming modules. One shortcoming of some of the original modules and their machine learned replacements is lack of adherence to conservation principles that are essential to model performance. In this work, we derive a mathematical framework for machine learned replacements that conserves properties, say mass, atoms, or energy, to machine precision. This framework can be used to develop machine learned operator replacements in environmental models.

## 1 Introduction

Complex systems require large models that simulate the wide range of physical and chemical properties that govern their performance. In the air quality realm, models include CMAQ (Foley, Roselle et al. 2010), CAMx (Yarwood, Morris et al. 2007), WRF-Chem (Grell, Peckham et al. 2005) and GEOS-Chem (Eastham, Weisenstein et al. 2014). In the climate change arena, models include HadCM3 (Jones, Gregory et al. 2005), GFDL CM2 (Delworth, Rosati et al. 2012), ARPEGE-Climat (Somot, Sevault et al. 2008), CESM (Kay, Deser et al. 2015), and E3SM (Golaz, Caldwell et al. 2019). These models employ operator splitting, also called the method of fractional steps (Janenko 1971), in their structure, so that each module can be

tasked with representing one or a small number of physical and/or chemical processes. This modular structure enhances model maintenance and sustainability while enabling diverse physical and chemical processes to interact. Each module is tasked with simulating its processes over a fixed period of time, each module called in turn until they have all returned their results. Usually,
the computational performance of these models is governed by one or two modules that consume the vast majority of the computer resources. In air quality models, this is usually the photochemistry and/or aerosol dynamics modules. In climate models, this is usually the radiative energy transport module.

Machine learning has been used to improve the computational efficiency of modules in atmospheric models for decades
(Potukuchi and Wexler 1997). As machine learning algorithms have improved, these efforts have matured (Hsieh 2009, Kelp, Tessum et al. 2018, Rasp, Pritchard et al. 2018, Keller and Evans 2019, Pal, Mahajan et al. 2019). But the effort to replace physical and chemical operators with machine learned modules is challenging because small systematic errors can build. For instance, an 0.1% error over a 1 hour time step could lead to a 72% error after a month of simulation. This problem is compounded if the replacement module does not conserve quantities that are essential to model accuracy, such as atoms in a
photochemical module, molecules and mass in an aerosol dynamics module, or energy in a radiative transfer module.

Recent efforts at developing and using machine learned replacement modules has focused on memorizing how the quantities change. If instead, we focus on how the fluxes between quantities change, we can guarantee adherence to conservation principles to machine precision. In photochemical modules, the fluxes are how atoms move between chemical species as
reactions progress. In aerosol dynamics, the fluxes are the condensation/evaporation or coagulation processes that move material between the gas and particle phases or between particle sizes. In radiative transfer, the fluxes are the energy movements between spatial domains.

In this work, we derive a mathematical framework that enables the use of machine learning tools to memorize these fluxes.
We focus this work on atmospheric photochemistry and provide an example for a simple photochemical reaction mechanism, because the number of species and the complexity of the problem exercises many aspects of the framework.



## 2 Glossary of Symbols

$C_i(t)$ – concentration at time $t$

$C_i(t + \Delta t)$ – concentration at time $t + \Delta t$

$\Delta C_i \equiv C_i(t + \Delta t) - C_i(t)$

$i = 1, n$ – the number of molecular species

$\Delta t$ – operator splitting time step

$R_j(t)$ – contribution to $\Delta C_i$ from each reaction

$S_j(t) = \int_t^{t+\Delta t} R_j(\tau) d\tau$

$j = 1, m$ – the number of reactions, $m > n$

$A$ – stoichiometry matrix relating $\Delta C_i$ to $S_j$, sparse, most element values are 0, 1 or -1

$A^G$ – generalized inverse of $A$

$A_S^G$ – constrained generalized inverse of $A$


## 3 Derivation of the Framework for Photochemistry

In general, the atmospheric chemistry operator solves

$$\frac{\partial C}{\partial t} = F(C, T, RH, \text{actinic flux, stability, etc.}) \tag{1}$$

where $C$ is a vector containing the current concentration of the chemical species, $T$ is temperature and $RH$ is the relative humidity. The right hand side can be written as

$$F = AR \tag{2}$$

where $A$ is a matrix describing the stoichiometry and $R$ is a vector of reaction. The form of the right hand side assures mass balance because it is compose of reactions that destroy one species while creating one or more other ones, all in balance,

described by $A$. The $R$ terms take forms such as $kC_iC_j$ where $k$ is the rate constant for a reaction between species, $JC_i$ where $J$ is the photolysis reaction rate, or $k(C_i(x_1) - C_i(x_2))$ where $k$ is a diffusion or mass transport rate constant between two spatial locations or between the gas and particle phases.

In the method of fractional steps, all modules integrate their equations forward for a fixed time step, $\Delta t$, that we call the

operator splitting time step. Combining these two equations and integrating gives

$$\Delta C_i = \sum_j A_{i,j} \int_t^{t+\Delta t} R_j(t) dt = \sum_j A_{i,j} S_j \tag{3a}$$





Or in matrix form

$$\Delta C = AS \tag{3b}$$

where $S_j = \int_t^{t+\Delta t} R_j(t)dt$. $S_j$ is the flux integral. For atmospheric photochemistry, it is the flux of atoms between molecules. For aerosol dynamics, it is the flux of molecules condensing on or evaporating from particles or the flux of small particles coagulating on large particles. For radiative transfer, $S_j$ is the energy between spatial coordinates. We are able to pull the $A$ out of the integral if it is a constant, which is usually the case or can be approximated as such.

Using machine learning tools to learn the relationship

$$S = S(C, T, RH, \text{actinic flux}, \text{stability}, \text{etc.}) \tag{4}$$

has advantages over memorizing a concentration-concentration relationship because:

    a.   The formulation in Eq. (3) conserves mass.

    b.   The $R$ terms are simpler to memorize because they do not contain the complexity in $A$.

c.   There are fewer concentrations directly influencing $S$ than $C$ so the machine learning algorithm should be simpler.

The difficulty resides in developing the training and testing sets needed to train and test the machine learning algorithm corresponding to Eq. (4). In principle, we can run a model many times, generate a data set, and then learn that data using machine learning techniques. That is, we can run many models that integrate Eq. (1) to find the relationship between concentrations at two time steps to develop our machine learning training set. But such models do not provide the value of $S$ and since the chemical system is stiff the integrators make many complex calls to calculate the right hand side of Eq. (1) to integrate it. Another way of saying this is that the $\Delta C$ is easily available from the models but the $S$ is not.

If we have many sets of $\Delta C$ values, in principle we can invert Eq. (3b) to obtain the corresponding $S$ values. The difficulty with this approach is that there are more elements of $S$ than $\Delta C$, so a conventional inverse cannot be applied. Instead, we employ the generalized inverse of $A$ to obtain $S$ via the relationship

$$S = A^G \Delta C \tag{5}$$

where $A^G$ is the generalized inverse of $A$. Since in general $A$ is not square and even if it is square, it may be singular, there are an infinite number of generalized inverses $A^G$ which means that given values for $\Delta C$, Eq. (5) will not give reliable values for $S$.

Given sufficient constraints, $A^G$ will be unique and provide the desired values of $S$ that are needed to develop a machine learning training set. Ben-Israel and Greville (Ben-Israel and Greville 2003) show that the inverse can be unique if the





solutions, $S$, are restricted to lie in a subspace that defines the "legal" solutions and these restrictions are sufficiently

constraining. The constrained generalized inverse of $A$ the produces solutions, $S$, that lie in the legal subspace defined by good examples of solutions $S$ is given by

$$A_S^G = P_S(AP_S)^G \tag{6}$$

where $A_S^G$ is the generalized inverse of $A$ restricted to the subspace of all possible solutions by the projection $P_S$, which in turn is defined by a set of basis vectors that define the subspace. Before we discuss obtaining the basis vectors, we first need to

discuss how to obtain the projection, $P_S$.

Assume for the moment that we have the basis vectors $S_k$. We concatenate them (column-wise) to form the matrix $U$:

$$U = \langle S_1 | S_2 | \ldots | S_k \rangle \tag{7}$$

The projection onto the subspace defined by these basis vectors $S_k$ and the matrix $U$ is then (Mukhopadhyay 2014):

$$P_S = U(U^+ U)^{-1} U^+ \tag{8}$$

where $U^+$ is the transpose of $U$.

Atmospheric chemistry problems are stiff so the $U^+U$ may be ill-conditioned. One source of this ill conditioning which also can hamper machine learning tools is that the concentrations are often orders of magnitude apart. The modules use actual

concentrations to make the mechanism easier to understand and debug. Normalizing the concentrations helps with both learning and stiffness/ill conditioning. Since the $S$ vectors describe the subspace where the solutions must reside, their magnitude does not matter, just their direction. So we normalize the $S$ vectors by dividing by the average of the non-zero values. Mathematically, we form a diagonal square matrix, $N_S$, with the averages on the diagonal and calculate the normalized $S$ with

$$S_{norm} = N_S^{-1} S \tag{9}$$

Since $N_S$ is diagonal, the inverse is simply the reciprocal of each diagonal element. The $\Delta C$ values are recovered from the $S_{norm}$ values via

$$\Delta C = A N_S S_{norm} \tag{10}$$

Atmospheric chemistry problems are also high dimensional. Typical air quality models may have 100 to 200 chemical species

and since the vertical column mixing time scale is similar to the slower time scales of the chemistry, some models solve the vertical transport and chemistry simultaneously. Since typical air quality models have 10 to 20 vertical cells, the dimension of the problem is 1,000 to 4,000. Even though the inverse $(U^+U)^{-1}$ only has to be calculated once, this inversion may be intractable. Providing that the condition number of $U^+U$ is not too large, Gram-Schmidt orthonormalization can be performed





on the $S_i$ vectors before carrying out Eqs. (7) and (8) in which case they will describe the same subspace but now the matrix
$U^+U$ will be the identity matrix which is its own inverse.

Now let us return to the question of how to find the basis vectors that define the "legal" subspace of S. These can be developed by solving Eq. (1) using Euler's method, in which case Eq. (3) becomes

$$\Delta C_i \sim A_{i,k} R_k(t) \Delta t \sim A_{i,k} S_k \tag{11}$$

That is

$$S_k \sim R_k(t) \Delta t \tag{12}$$

The value of $\Delta t$ does not matter since it just changes the length of $S_k$ not its direction and therefore not its value in describing the subspace. The original module that calculates $R_k$ can be run many times under many conditions to generate a set of $S_k$ vectors that span the subspace. Then Locality Preserving Projections (LPP), Principal Component Analysis (PCA) or another similar algorithm can be used to find a minimum set of vectors that define the subspace.

**4 Solution Procedure for a Photochemical Module**

1. Determine which species are active in the photochemical mechanism. That is, not the steady state or build-up species.
2. From the mechanism, extract the A matrix for these species.
3. Using a representative set of atmospheric concentrations, T, RH, and actinic flux, use Eq. (10) and the photochemical module to generate data that match values of $\Delta C$ and $S$ for many values of $C$, T, RH, and actinic flux for the models operator splitting time step.
4. Normalize the $S$ vectors by dividing each by the average of its nonzero elements. Use these averages to form the $N_S$ matrix, which relates $S$ to $S_{norm}$ via Eq. (9).
5. Use the $S_{norm}$ vectors and Eq. (7) to form the $U$ matrix and then the $U^+U$ matrix. What is the condition number of the $U^+U$ matrix? If the system is large and not ill-conditioned, use Gram-Schmidt orthonormalization on the $S$ vectors before calculating $U$ and $U^+U$, in which case $U^+U$ should be an identity matrix or a subset of one.
6. Use Eq. (8) to calculate $P_S$.
7. Use Eq. (6) to calculate the constrained generalized inverse $A_S^G$.
8. Use Eq. (5) to calculate values of $S$ from the values of $\Delta C$.
9. Compare the values of $S$ obtained from steps 3 and 7 to make sure they are very similar using the dot product to calculate the angle between them. If they are, we have a good $A_S^G$.
10. Use neural networks or another machine learning algorithm to memorize the $S(C)$ relationship obtained using (a) Eq. (5), and (b) many runs of the mechanism for a wide range of $C$, T, RH, and actinic flux values.
11. Replace the mechanism with the neural network to calculate $S(C)$ and Eq. (3b) to march forward.





12. Clock the speed improvement.

13. Calculate standard measures of performance such as mass balance, bias, and error compared to runs using the complete mechanism.

## 5 Photochemical Mechanism

We tested the methods described above on the following very simplified set of photochemical reactions used by Dr. Kleeman at the University of California, Davis when teaching the modeling of atmospheric photochemistry. Although this mechanism is abbreviated, it contains the essential components of all atmospheric photochemical mechanisms related to ozone formation: NOx chemistry, VOC chemistry, formation of peroxy radicals from VOC chemistry that then react with NO to form NO2 and OH, both of which may react to terminate.

The 10 reactions are given in Table 1. The oxygen atom and hydroxyl radical are assumed to be in steady state so there are 6 active species, which are listed in Table 2.

| Table 1: Reaction Mechanism | |
|---|---|
| Reaction | Reaction Number |
| $NO_2 + hv \rightarrow NO + O$ | R1 |
| $O + O_2 \rightarrow O_3$ | R2 |
| $O_3 + NO \rightarrow NO_2 + O2$ | R3 |
| $HCHO + hv \rightarrow 2\ HO_2^{.} + CO$ | R4 |
| $HCHO + hv \rightarrow H2 + CO$ | R5 |
| $HCHO + HO^{.} \rightarrow HO_2^{.} + CO + H_2O$ | R6 |
| $HO_2^{.} + NO \rightarrow OH^{.} + NO_2$ | R7 |
| $OH^{.} + NO_2 \rightarrow HNO_3$ | R8 |
| $HO_2H + hv \rightarrow 2\ HO^{.}$ | R9 |
| $HO_2H + HO^{.} \rightarrow H_2O + HO_2^{.}$ | R10 |





| Table 2. Active Species |
|---|
| $O_3$ |
| NO |
| $NO_2$ |
| HCHO |
| $HO_2\cdot$ |
| $HO_2H$ |

The resulting $A$ matrix represents the stoichiometry of the reactions where the rows correspond to each species and the columns to each reaction:

$$A = \begin{bmatrix} & R1 & R2 & R3 & R4 & R5 & R6 & R7 & R8 & R9 & R10 \\ O_3 & 0 & 1 & -1 & 0 & 0 & 0 & 0 & 0 & 0 & 0 \\ NO & 1 & 0 & -1 & 0 & 0 & 0 & -1 & 0 & 0 & 0 \\ NO_2 & -1 & 0 & 1 & 0 & 0 & 0 & 1 & -1 & 0 & 0 \\ HCHO & 0 & 0 & 0 & -1 & -1 & -1 & 0 & 0 & 0 & 0 \\ HO_2 & 0 & 0 & 0 & 2 & 0 & 1 & -1 & 0 & 0 & 1 \\ HO_2H & 0 & 0 & 0 & 0 & 0 & 0 & 0 & 0 & -1 & -1 \end{bmatrix}$$

As in prior efforts (Kelp, Tessum et al. 2018, Keller and Evans 2019), we employed a box model in Julia to generate 60 independent days of output for both $\Delta C$ and $S$, recording data every 6 minutes. We are interested in the set of $S$ vectors that form a basis describing the subspace that contains the desired $S$ vectors. First, the transformation in Eq. (9) is performed to normalize the sample $S$ vectors. In this example, we use LPP (Locality Preserving Projections) (He and Niyogi 2004), which is similar to PCA (Principle Component Analysis) but more robust for this application. Here LPP yields a basis set of 7 vectors, which form the columns of the $U$ matrix:

$$U = \begin{bmatrix} -0.6869 & 0.1334 & -0.2068 & -0.1461 & 0.0867 & -0.3715 & 0.4761 \\ -0.6869 & 0.1334 & -0.2068 & -0.1461 & 0.0867 & -0.3715 & 0.4761 \\ -0.1877 & -0.1444 & 0.0260 & 0.1967 & -0.0540 & 0.5443 & -0.7353 \\ 0.0406 & -0.5849 & -0.0080 & -0.2426 & 0.1194 & 0.1747 & 0.0027 \\ 0.0411 & -0.5911 & -0.0081 & -0.2452 & 0.1207 & 0.1765 & 0.0027 \\ -0.0202 & -0.2414 & -0.0069 & -0.0875 & 0.0844 & -0.3942 & -0.0570 \\ 0.0149 & -0.2555 & 0.0154 & 0.0759 & -0.1568 & -0.2242 & -0.0519 \\ 0.1063 & -0.1359 & 0.0844 & 0.4800 & -0.7829 & 0.4002 & -0.0066 \\ -0.0670 & -0.0762 & 0.9337 & 0.0057 & 0.0069 & -0.0136 & -0.0012 \\ -0.0354 & -0.3227 & -0.1856 & 0.7455 & 0.5554 & 0.0107 & -0.0013 \end{bmatrix}$$





The condition number of $U^+U$ is approximately 12. This is several orders of magnitude smaller than the same problem but without the $N_S$ transformation of Eq. (9) where the condition number was 1888. High condition numbers mean that the matrix inversion is problematic at best. The condition number of 12 after the $N_S$ transformation ensures that the inversion needed to

make the projection $P_S$ in Eq. (8) is numerically tractable.

The resulting symmetric block diagonal projection $P_S$ is equal to:

$$P_S = \begin{bmatrix} 0.500 & 0.500 & 0.000 & 0.000 & 0.000 & 0.000 & 0.000 & 0.000 & 0.000 & 0.000 \\ 0.500 & 0.500 & 0.000 & 0.000 & 0.000 & 0.000 & 0.000 & 0.000 & 0.000 & 0.000 \\ 0.000 & 0.000 & 1.000 & 0.000 & 0.000 & 0.000 & 0.000 & 0.000 & 0.000 & 0.000 \\ 0.000 & 0.000 & 0.000 & 0.495 & 0.500 & 0.000 & 0.000 & 0.000 & 0.000 & 0.000 \\ 0.000 & 0.000 & 0.000 & 0.500 & 0.505 & 0.000 & 0.000 & 0.000 & 0.000 & 0.000 \\ 0.000 & 0.000 & 0.000 & 0.000 & 0.000 & 0.587 & 0.471 & -0.142 & 0.001 & -0.005 \\ 0.000 & 0.000 & 0.000 & 0.000 & 0.000 & 0.471 & 0.462 & 0.163 & -0.001 & 0.006 \\ 0.000 & 0.000 & 0.000 & 0.000 & 0.000 & -0.142 & 0.163 & 0.951 & 0.000 & -0.002 \\ 0.000 & 0.000 & 0.000 & 0.000 & 0.000 & 0.001 & -0.001 & 0.000 & 1.000 & 0.000 \\ 0.000 & 0.000 & 0.000 & 0.000 & 0.000 & -0.005 & 0.006 & -0.002 & 0.000 & 1.000 \end{bmatrix}$$


And Eq. (6) gives us

$$A_S^G = \begin{bmatrix} 2.70E1 & 0.0000 & 0.0000 & 0.0000 & 0.0000 & 0.0000 \\ 2.70E1 & 0.0000 & 0.0000 & 0.0000 & 0.0000 & 0.0000 \\ -4.18E1 & 0.0000 & 0.0000 & 0.0000 & 0.0000 & 0.0000 \\ 3.63E3 & -5.45E3 & -1.82E3 & 3.63E3 & 5.45E3 & 3.63E3 \\ 3.67E3 & -5.51E3 & -1.84E3 & 3.67E3 & 5.51E3 & 3.67E3 \\ -2.84E3 & 4.26E3 & 1.42E3 & -3.45E3 & -4.26E3 & -2.84E3 \\ 5.37E2 & -5.37E2 & 0.0000 & 0.0000 & 0.0000 & 0.0000 \\ 0.0000 & -1.78E3 & -1.78E3 & 0.0000 & 0.0000 & 0.0000 \\ -9.24E5 & 1.14E6 & 2.16E5 & -9.24E5 & -1.14E6 & -9.24E5 \\ 9.81E4 & -1.21E5 & -2.29E4 & 9.81E4 & 1.21E5 & 4.58E4 \end{bmatrix}$$

Since $U$ and $A_S^G$ have 7 and 6 independent columns, respectively, but 10 rows, and the row rank is equal to the column rank, there must be linearly dependent rows. One manifestation of this is that the first two rows of $U$ and $A_S^G$ are identical, or nearly so. The $S$ values computed from $A_S^G$ may not be identical to the original $S$ corresponding to the $\Delta C$ values. However, all $S$ values calculated from Eq. (5) using the above $A_S^G$ are "legal": in other words, within the subspace defined by the basis set $U$. Furthermore, the inverse $A_S^G$ by definition satisfies $AA_S^G = I$, so that even if a calculated $S$ is not identical to the $S$ from the

original box model output, it can be used in Eq. (3b) to return a $\Delta C$ identical to that of the box model output.





## 6 Conclusions

Large models of the environment require solution of large systems of equations over long periods of time. These models consume vast quantities of computational resources so approximations are necessarily employed so that the models are computationally tractable. Machine learning tools can be used to dramatically improve the speed of these models enabling more faithful representation of the physics and chemistry while also improving runtime performance. But this field is in its infancy. To help facilitate the use of machine learning tools in these environmental models, we have developed a framework that (a) enables machine learning algorithms to learn flux terms assuring that conservation principles dictated by the physics

and chemistry are adhered to and (b) allow parameters easily calculated by geophysical models to be used to back calculate these flux terms that can then be used to train the machine learning algorithm of choice. Applications of this framework in environmental model include any process where conservation principles apply such as conservation of atoms in chemical reactions, conservation of molecules during phase change, and conservation of energy in say radiative transfer calculations.

**Author Contribution**

ASW initiated the project and is responsible for the conceptualization of the mass balancing framework. ASW and POS contributed to the formal analysis, including the generalized inverse and preconditioning approach. POS developed the model code using Julia and MATLAB.

**Competing Interests**

The authors declare that they have no conflict of interest.

**Code Availability**

The most current version of the MATLAB script used to generate $A_S^G$ and the projection is available at

https://doi.org/10.5281/zenodo.3712457, and the input data at https://doi.org/10.5281/zenodo.3733502. The exact version of the script used to produce the results used in this paper is named *GenerateAG.m* and is archived on Zenodo (https://doi.org/10.5281/zenodo.3733594, Sturm P.O. and A.S Wexler 2020). The input files required for this script, as well as the Julia mechanism, are available on Zenodo as *S.txt* and *delC.txt* (https://doi.org/10.5281/zenodo.3733503, Sturm 2020). Both the restricted inverse script and the input data are available under a Creative Commons Attribution 4.0 International

license.

**Acknowledgements**

Dr. Michael Kleeman at University of California, Davis contributed the photochemical mechanism which was modified to create the data needed to calculate the restricted inverse.

The LPP MATLAB program was written by Dr. Deng Cai at the College of Computer Science Zhejiang University, China. Both the Fortran and LPP programs were essential in calculating the restricted inverse described in this paper.



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
