# Peer review of "A Mass- and Energy-Conserving Framework for Using Machine Learning to Speed Computations: A Photochemistry Example"

_Geoscientific Model Development, 2020_

## Referee Comment (RC1) · Anonymous Referee #1 · 29 May 2020

This paper covers the process for creating a mass-conserving ML framework for a photochemistry code embedded in a larger model. This process is described in detail and appears to be reproducible, and the process is both useful and robust. Given the content of the paper, it seems the title should be altered to specifically mention photochemistry.

Unless I have misread, the crux of this paper's approach to conserving mass or energy seems to be predicting terms that naturally give rise to conservation in the underlying discretization being modeled (e.g., fluxes for radiation or the time-integrated reaction vector, which is re-distributed in a conserving way by the stoichiometry matrix). Seemingly one could also create a conservative ML surrogate for, say, a Finite-Volume or Discontinuous Galerkin discretization by predicting the fluxes rather than only the tendencies. This idea is useful but not necessarily novel in itself.

What is very novel is how one extracts a learnable quantity from the more readily available variables in the photochemistry application, a process that took quite a bit of careful attention. This is why I think the title might be better suited to the more novel demonstration of the concept in a complicated application. As I was reading, I was waiting to read about an example using radiation as well (mainly because of the title's generality), even though on a second pass, it was not stated that a radiation example would be given.

While the focus of this paper is machine precision conservation in a ML framework, since a ML surrogate model was indeed produced, it would be helpful to get a sense of the accuracy that was obtained compared to the original codebase for a set of representative examples one might encounter in a realistic model.

---

## Referee Comment (RC2) · Anonymous Referee #2 · 13 Jun 2020

The current study uses an energy and mass conservation framework in a machine learning module to speed up the atmospheric chemistry computations in large scale simulations. The authors have suggested the use of machine learning tools to memorize the fluxes of change in quantities instead of change in quantities themselves which guarantees adhesion to conservation principles. The paper is well written and is within the scope of the journal, I have some minor suggestions to be included.

1. The authors in line 105 mentions that models usually don't give the flux "S" as output yet while providing the solution for a photochemical module in line 161, the authors in step 3 talk of generating the $\Delta C$ and S. How do the authors get the "S" value here?

2. The authors are suggested to clearly mention and segregate the steps taken to generate the test and train data and then the steps from which machine learning tools are used to learn the relationship of S with other factors in section 3, 4 and 5. In short the authors should clearly mention the steps to be done using Chemical Transport Models (CTM) and the steps where the machine learning algorithm is used to determine the relationship of S with other factors.

3. The authors doesn't mention regrading the errors in approximating equations 11 and 12. Any suggestions to reduce the same?

4. The authors mention 2 problems which may occur a) regarding the stiffness of atmospheric chemistry problems b) regarding high dimensionality of atmospheric chemistry problems. Can the authors add more explanation with suitable examples for the problem as well as the suggested solution for better understanding?

---

## Author Comment (AC1) · 10 Jul 2020

The comment was uploaded in the form of a supplement:
https://gmd.copernicus.org/preprints/gmd-2020-83/gmd-2020-83-AC1-supplement.pdf

---

## Author Response (AR1)

Author Reply Obin Sturm and Anthony Wexler

We would like to thank the reviewers and editor for their time and contributions to improving this work. Below, we respond to comments for both of the reviews. Our responses are in bold.

**Anonymous Referee #1**

This paper covers the process for creating a mass-conserving ML framework for a photochemistry code embedded in a larger model. This process is described in detail and appears to be reproducible, and the process is both useful and robust. Given the content of the paper, it seems the title should be altered to specifically mention photochemistry.

**Thanks for your review. We agree that the example should be mentioned in the title – we've updated the title accordingly.**

What is very novel is how one extracts a learnable quantity from the more readily available variables in the photochemistry application, a process that took quite a bit of careful attention. This is why I think the title might be better suited to the more novel demonstration of the concept in a complicated application. As I was reading, I was waiting to read about an example using radiation as well (mainly because of the title's generality), even though on a second pass, it was not stated that a radiation example would be given.

The main focus of this paper is the derivation of the conservation matrix and then the extraction of the learnable quality S. We changed the title to make clear that the numerical example is from a photochemistry application. We mention examples of how this approach can be generalized to other applications in lines 45, 46, 54-57 on page 2, as well as in the derivation on page 3, lines 85-87, and page 4, lines 95-98.

While the focus of this paper is machine precision conservation in a ML framework, since a ML surrogate model was indeed produced, it would be helpful to get a sense of the accuracy that was obtained compared to the original codebase for a set of representative examples one might encounter in a realistic model.

We agree that would be helpful! We are currently working on the application of this to two different realistic mechanisms (photochemistry and aerosol dynamics), and hope to describe that work in a different manuscript type as outlined by GMD – either a model evaluation or model experiment description paper. As the main emphasis of this paper is the derivation of the method, as well as a look into improving robustness when encountering potential numerical issues, we decided this paper should be included in the GMD manuscript type development and technical paper, focusing on the introduction of the novel approach.

**Anonymous Referee #2**

The current study uses an energy and mass conservation framework in a machine learning module to speed up the atmospheric chemistry computations in large scale simulations. The authors have suggested the use of machine learning tools to memorize the fluxes of change in quantities instead of change in quantities themselves which guarantees adhesion to conservation principles. The paper is well written and is within the scope of the journal, I have some minor suggestions to be included.

**Thank you for your review and comments. We are glad to hear your positive recommendation.**

1. The authors in line 105 mentions that models usually don't give the flux "S" as output yet while providing the solution for a photochemical module in line 161, the authors in step 3 talk of generating the  $\Delta$ C and S. How do the authors get the "S" value here?

**The section on page 6, lines 160-168 details how to generate values that span the space of realistic S values.**

2. The authors are suggested to clearly mention and segregate the steps taken to generate the test and train data and then the steps from which machine learning tools are used to learn the relationship of S with other factors in section 3, 4 and 5. In short the authors should clearly mention the steps to be done using Chemical Transport Models (CTM) and the steps where the machine learning algorithm is used to determine the relationship of S with other factors.

We have added a few sentences at the beginning of section 4 on page 7 to clarify which steps in the process are handled by the algorithm described in this paper and which steps follow. The algorithm described here can be used on any problem where conservation principles need to be adhered to and it does not restrict the machine learning tools to be deployed solving that problem.

3. The authors doesn't mention regrading the errors in approximating equations 11 and 12. Any suggestions to reduce the same?

**The purpose of these equations is not to approximate specific fluxes, but rather to obtain a representative subspace of fluxes for use in generating the restricted inverse. The accuracy of each flux direction will be dictated by the original mechanism.**

4. The authors mention 2 problems which may occur a) regarding the stiffness of atmospheric chemistry problems b) regarding high dimensionality of atmospheric chemistry problems. Can the authors add more explanation with suitable examples for the problem as well as the suggested solution for better understanding?

Thanks for pointing this out. We have added a sentence in the derivation section on page 5, line 144, as well as an additional sentence on page 9, lines 226, directly after the U matrix, that explicitly link stiffness of the problem with condition numbers and invertibility. The matrices  $N_S$ , U and reported condition number for  $U^+U$  are examples of successful handling of numerical issues arising from stiffness.

Within the scope of this paper we are not able to provide a numerical example that handles high dimensionality, since the example mechanism in this paper has low dimensionality. However, on page 6, lines 151 - 158, we include a brief discussion on a possible approach to making high dimensional inversions computationally tractable, to indicate that other numerical strategies might be needed with larger systems.

**Additional Changes**

In response to the suggestions from both referees as well as colleagues who have reviewed this work, we have made the following minor additions. These are intended to improve accessibility and context of this manuscript.

- 1) A few sentences have been added in the introduction, on page 2, lines 49-53, to reference some other recent work involving physical constraints in machine learning.
- 2) There are cases that might not require construction of the restricted inverse, and these are briefly discussed on lines 118-123 in section 3.

[revised manuscript text omitted]

230  $P_S$  in Eq. (8) is numerically tractable.

|         | r0.500      | 0.500 | 0.000 | 0.000 | 0.000 | 0.000  | 0.000  | 0.000  | 0.000  | ך 0.000 |
|---------|-------------|-------|-------|-------|-------|--------|--------|--------|--------|---------|
|         | 0.500       | 0.500 | 0.000 | 0.000 | 0.000 | 0.000  | 0.000  | 0.000  | 0.000  | 0.000   |
|         | 0.000       | 0.000 | 1.000 | 0.000 | 0.000 | 0.000  | 0.000  | 0.000  | 0.000  | 0.000   |
|         | 0.000       | 0.000 | 0.000 | 0.495 | 0.500 | 0.000  | 0.000  | 0.000  | 0.000  | 0.000   |
| ת       | _ 0.000     | 0.000 | 0.000 | 0.500 | 0.505 | 0.000  | 0.000  | 0.000  | 0.000  | 0.000   |
| $P_{S}$ | = 0.000     | 0.000 | 0.000 | 0.000 | 0.000 | 0.587  | 0.471  | -0.142 | 0.001  | -0.005  |
|         | 0.000       | 0.000 | 0.000 | 0.000 | 0.000 | 0.471  | 0.462  | 0.163  | -0.001 | 0.006   |
|         | 0.000       | 0.000 | 0.000 | 0.000 | 0.000 | -0.142 | 0.163  | 0.951  | 0.000  | -0.002  |
|         | 0.000       | 0.000 | 0.000 | 0.000 | 0.000 | 0.001  | -0.001 | 0.000  | 1.000  | 0.000   |
|         | $L_{0.000}$ | 0.000 | 0.000 | 0.000 | 0.000 | -0.005 | 0.006  | -0.002 | 0.000  | 1.000 J |

**And Eq. (6) gives us**

[revised manuscript text omitted]